# LEARNING TO WRITE BY LEARNING THE OBJECTIVE

## ABSTRACT

Recurrent Neural Networks (RNNs) are powerful autoregressive sequence models for learning prevalent patterns in natural language. Yet language generated by RNNs often shows several degenerate characteristics that are uncommon in human language; while fluent, RNN language production can be overly generic, repetitive, and even self-contradictory. We postulate that the objective function optimized by RNN language models, which amounts to the overall perplexity of a text, is not expressive enough to capture the abstract qualities of good generation such as Grice's Maxims. In this paper, we introduce a general learning framework that can construct a decoding objective better suited for generation. Starting with a generatively trained RNN language model, our framework learns to construct a substantially stronger generator by combining several discriminatively trained models that can collectively address the limitations of RNN generation. Human evaluation demonstrates that text generated by the resulting system is preferred over that of baselines by a large margin and significantly enhances the overall coherence, style, and information content of the generated text.

## 1 INTRODUCTION

Recurrent Neural Network (RNN) based language models such as Long Short-Term Memory Networks (LSTMs) (Hochreiter & Schmidhuber, 1997) and Gated Recurrent Units (GRUs) (Cho et al., 2014) have achieved enormous success across a variety of language tasks due to their ability to learn fluency patterns in natural language (Jozefowicz et al., 2016; Kim et al., 2016; Mikolov et al., 2010). When used as a generator, however, the quality of language generated from RNNs deviates drastically from that of human language. While fluent, RNN-produced language displays several degenerate characteristics, favoring generic and contentless output that tends to be repetitive and self-contradictory. These issues are especially prominent when RNNs are used for open-ended, long-form text generation, as illustrated in Figure 1.

RNNs model the conditional probability $P(x_t|x_1, ..., x_{t-1})$ of generating the next word $x_t$ given all previous words observed or generated. In theory, this conditional model should be able to learn all crucial aspects of human language production, for example, that we don't normally repeat the same content over and over. In practice, however, the learned conditional probability model often assigns higher probability to a repetitive, overly generic sentence than to higher quality sentences, as shown in Figure 1. We postulate that this is in part because the network architectures of RNN variants do not provide a strong enough inductive bias for the model to learn the complex communication goals pursued in human writing. In addition, long-term context easily gets lost as it is *explained away* in the presence of more immediately relevant short-term context (Yu et al., 2017), and as gradients diminish over a long sequence (Pascanu et al., 2013). Consequently, RNNs acquire relatively shallow and myopic patterns, which tend to only take advantage of a small fraction of the training set vocabulary Kiddon et al. (2016). RNNs are thus unable to generate language that matches the complexity and coherence of human generated text.

Several methods in the literature attempt to mitigate these issues. Overly simple and generic generation can be improved by using a diversity-boosting objective function (Shao et al., 2017; Vijayakumar et al., 2016). Repetitive generation can be reduced by prohibiting recurrence of the same trigrams as a hard rule (Paulus et al., 2017). Although such constraints form a partial solution, they

**All in all, I would highly recommend this hotel to anyone who wants to be in the heart of the action, and want to be in the heart of the action. If you want to be in the heart of the action, this is not the place for you. However, If you want to be in the middle of the action, this is the place to be.**

Figure 1: A Trip Advisor review generated by an RNN based LM trained on over a million reviews.

are generally too coarse and both penalize good behavior (e.g. reuse of an idiom) and fail to capture more complex bad behavior (e.g. paraphrasing of the same content again and again).

Hand tailoring rules is both time consuming and unstable across different generative scenarios, so we instead propose a general learning framework to construct a better decoding objective. Starting with a generatively trained RNN language model, our framework learns to construct a substantially stronger generator by combining several discriminatively trained models that can collectively address limitations of the base RNN generator. Our learning framework therefore generalizes over various existing modifications to the decoding objective. Our approach learns to overcome the particular limitations of the RNN generator directly by incorporating language generated from RNNs as negative samples to discriminatively train several companion models, each specializing in a different aspect of Grice's Maxims of communication (Grice et al. (1975)).

Empirical results demonstrate that our learning framework is highly effective in converting a generic RNN language model into a substantially stronger generator. Human evaluation confirms that language generated by our model is preferred over that of competitive baselines by a large margin and significantly enhances the overall coherence, style, and information content of the generated text.

## 2 BACKGROUND: GRICE'S MAXIMS

We motivate our learning framework using Grice's Maxims of communication (Grice et al., 1975):

**1. Quantity:**  *Make your contribution as informative as required, and no more than necessary.*

RNN generations tend to violate this maxim as they are often overly short and generic, and therefore less informative than desired. When encouraged to generate longer text, RNNs easily repeat themselves, which also works against conveying the right amount of information. This observation motivates the *length* and *repetition* models in §3.2.1 and §3.2.2.

**2. Quality:**  *Do not say what you believe to be false.*

When used for generation with long term context, RNNs often generate text that is self-contradictory. We propose an *entailment* model to address this problem in §3.2.3.

**3. Relation:**  *Be relevant.*

RNNs used for long generation can start digressing as the long-term context gets easily washed out. We address this problem by proposing two different *relevance* models in §3.2.4 and §3.2.5.

**4. Manner:**  *Avoid obscurity of expression. Avoid ambiguity. Be brief. Be orderly.*

As RNN generation favors highly probable but generic words and phrases, it lacks adequate style and specificity. We address this issue with the *lexical style* model in §3.2.6.

## 3 THE LEARNING FRAMEWORK

We propose a general learning framework for conditional language generation of sequence $\mathbf{y}$ given context $\mathbf{x}$. The decoding objective for generation takes the general form:

$$f(\mathbf{x}, \mathbf{y}) = \log(P_{\mathrm{lm}}(\mathbf{y}|\mathbf{x})) + \sum_k \lambda_k s_k(\mathbf{x}, \mathbf{y}). \tag{1}$$

This objective combines the RNN language model probability $P_{\mathrm{lm}}$ (§3.1) with a set of additional scores $s_k(\mathbf{x}, \mathbf{y})$ produced by discriminatively trained communication models (§3.2) and weighted

with learned mixture coefficients $\lambda_k$ (§3.3). This corresponds to a Product of Experts (PoE) model (Hinton, 2006) when the scores $s_k$ are log probabilities. Further model and hyperparameter details are given in appendix B.

Generation is performed using beam search (§3.4), scoring partially generated candidate generations $\mathbf{y}_{1:i}$ at each time step $i$. The RNN language model decomposes into per-word probabilities using the chain rule. However, in order to allow for more expressivity over long range context we do not require the discriminative model scores to factorize over the elements of $\mathbf{y}$, thereby addressing a key limitation of RNNs. More specifically, we use an estimated score $s'_k(\mathbf{x}, \mathbf{y}_{1:i})$ that can be computed for any prefix of $\mathbf{y} = \mathbf{y}_{1:n}$ to approximate the objective during beam search, such that $s'_k(\mathbf{x}, \mathbf{y}_{1:n}) = s_k(\mathbf{x}, \mathbf{y})$. The scoring functions are trained on prefixes of $\mathbf{y}$ (except where stated otherwise) to simulate their application to rank partial continuations during beam search.

## 3.1 BASE LANGUAGE MODEL

We train an RNN language model to estimate

$$\log P_{\text{lm}}(\mathbf{s}) = \sum_i \log P_{\text{lm}}(\mathbf{s}_i | \mathbf{s}_{1:i-1}). \tag{2}$$

The language model treats the context $\mathbf{x}$ and the continuation $\mathbf{y}$ as a single sequence $\mathbf{s}$, in contrast to sequence to sequence models which distinguish between them.

## 3.2 COMPOSITE COMMUNICATION MODELS

Next we introduce a set of models motivated by Grice's Maxims of communication. Each model is trained to discriminate between good and bad generation using a ranking objective, so that its model scores have the effect of re-ranking in the decoder. We vary the model parameterization and training examples to guide each model to focus on different aspects of Grice's Maxims. The classifier scores are interpreted as classification probabilities and added to the objective function as log probabilities.

Let $D = \{(\mathbf{x_1}, \mathbf{y_1}), \ldots (\mathbf{x_n}, \mathbf{y_n})\}$ be the set of training examples for conditional generation. $D_{\mathbf{x}}$ denote all contexts and $D_{\mathbf{y}}$ all continuations.

In all models the first layer embeds words into 300-dimensional vectors initialized with GloVe (Pennington et al., 2014) embeddings, which are fine-tuned during training unless stated otherwise. The dimensions of hidden layers are also 300. Let $e(w)$ be the word embedding of word $w$.

### 3.2.1 LENGTH MODEL

RNNs tend to bias toward shorter generation even with a highly expressive network structure with attention, thus length normalization is still a common practice (Wu et al., 2016). We use a geometrically decaying length reward. We tune the initial value and decaying rate on the final systems, and use the same value for all systems with the initial value and decay rate being 1 and 0.9 respectively. The score per a time step (for a partially generated completion) is thus

$$s_{\text{len}}(\mathbf{y}_{1:i}) = 0.9^i. \tag{3}$$

### 3.2.2 REPETITION MODEL

The goal of this model is to learn to distinguish between RNN-generated and gold continuations by exploiting our empirical observation that repetitions are more common in completions generated by the RNN language model. We quantify the claim that samples from the base RNN is an appropriate source of negative examples for repetition detection on our datasets in §4.1. However, we do not want to completely discourage repetition, as words sometimes do recur in English. Thus we model natural levels of repetition as follows.

First a score $d_i$ is computed for each token in the continuation $\mathbf{y}$ based on pairwise cosine similarity of word embeddings within a fixed window of the previous $k$ words:

$$d_i = \max_{j=i-k\ldots i-1} (\text{CosSim}(e(y_i), e(y_j))). \tag{4}$$

This score can be interpreted as the repetition "strength" of the sequence at position $i$, where $d_i = 1$ for all positions at which a word is repeated within $k + 1$ words of its previous use.

The score of the entire continuation is then defined as

$$s_{\text{rep}}(\mathbf{y}) = \sigma(\mathbf{w}_r^\top \text{RNN}(\mathbf{d})), \tag{5}$$

where $\text{RNN}(\mathbf{d})$ is the final state of a unidirectional RNN ran over the similarity scores and $\mathbf{w}_r$ is a learned vector. The model is trained to maximize binary cross-entropy, with gold continuations as positive examples and samples from the base RNN as negative examples:

$$L_{\text{rep}} = \sum_{\mathbf{y} \in D_{\mathbf{y}}} \log s_{\text{rep}}(\mathbf{y}) + \sum_{\mathbf{y}' \sim P_{\text{lm}}(\mathbf{y}')} \log(1 - s_{\text{rep}}(\mathbf{y}')). \tag{6}$$

Word embeddings are kept fixed during training for this model.

### 3.2.3 ENTAILMENT MODEL

The goal of the entailment model is to guide the generator to neither contradict its own past generation (the maxim of Quality) nor state something that readily follows from the context (the maxim of Quantity). The latter case is driven by the RNNs habit of paraphrasing itself (alongside its tendency for more direct repetitions, which is captured by the repetition model). We train a classifier using the MultiSNLI dataset (Williams et al., 2017) that takes two sentences $a$ and $b$ as input and predicts the relation between them as either *contradiction*, *entailment* or *neutral*. We use the score of the *neutral* class probability of the sentence pair in order to discourage both contradiction and entailment.

We use a continuous bag-of-words model similar to previously reported NLI baselines (Mou et al., 2016). Sentence representations are obtained by summing word embeddings (which are fine-tuned during training), such that $r(\mathbf{a}) = \sum_i e(a_i)$ and $r(\mathbf{b}) = \sum_i e(b_i)$. An MLP classifier with a single `tanh` hidden layer takes as input concatenations of the two sentence embeddings, along with their element-wise difference and multiplication; the output is a three-way softmax. The log probability of the neural class is $t(\mathbf{a}, \mathbf{b})$. The classifier obtains $63\%$ validation accuracy.

In contrast to our other communication models, this classifier cannot be applied directly to the full context and continuation sequences it is scoring. Instead every completed sentence in the continuation should be scored against all preceding sentences in both the context and continuation. Let $S(\mathbf{y}_{1:i})$ be the set of complete sentences in $\mathbf{y}_{1:i}$, and $S_{-1}(\mathbf{y}_{1:i})$ the last complete sentence. We compute the entailment score of $S_{-1}(\mathbf{y}_{1:i})$ against all preceding sentences in $\mathbf{x}$ and $\mathbf{y}$, and use the score of the sentence-pair for which we have the least confidence that entailment is neutral:

$$s_{\text{entail}}(\mathbf{x}, \mathbf{y}) = min_{\mathbf{a} \in S(\mathbf{x}) \cup S_{-1}(\mathbf{y})} t(\mathbf{a}, S_{-1}(\mathbf{y})). \tag{7}$$

In contrast to our other models, the score this model returns for any given continuation only corresponds to a subsequence of the continuation, but as we will see below (§3.4) due to the way generation is performed with beam search this score will not be accumulated across sentences.

### 3.2.4 RELEVANCE MODEL

The purpose of the relevance model is to predict whether the content of a candidate continuation is relevant to the given context. We train the model to distinguish between true continuations and random continuations sampled from other (human-written) endings in the corpus, conditioned on the given context. A single convolutional layer is applied to both the context and candidate embedding sequences. The scoring function is defined as

$$s_{\text{rel}} = \mathbf{w}_l^T \cdot (\text{maxpool}(\text{conv}(e(\mathbf{x}))) \circ \text{maxpool}(\text{conv}(e(\mathbf{y})))), \tag{8}$$

where 1D maxpooling is performed over each sequence to obtain a vector representing its most important semantic dimensions. Element-wise multiplication of the context and continuation vectors will amplify similarities between the two.

We optimize the following ranking log likelihood:

$$L_{\text{rel}} = \sum_{j=1}^{k} \sum_{(\mathbf{x}, \mathbf{y}_t) \in D, \mathbf{y}_r \sim D_{\mathbf{y}}} \log \sigma(s_{\text{rel}}(\mathbf{x}, \mathbf{y}_t) - s_{\text{rel}}(\mathbf{x}, \mathbf{y}_r)), \tag{9}$$

i.e., the probability of the true ending $\mathbf{y}_t$ receiving a higher score than the random ending $\mathbf{y}_r$. For each context and true continuation we extract $k = 5$ randomly selected endings as negative training examples. The model ranks true endings higher than randomly selected ones with $85\%$ accuracy on the validation set. The trained scores do not correspond to probabilities but we scale with the logistic (sigmoid) function for compatibility with other scoring modules.

### 3.2.5 WORKING VOCABULARY MODEL

Given even a fragment of context (e.g. "We were excited about our beach getaway...") certain topics become more relevant (e.g. "swimming", "relax", "romantic"). To capture this intuition, we predict a *centroid* in the embedding space from the context, which describes a neighborhood of the embedding space we expect the continuation to intersect. The score is computed as

$$s_{\text{voc}}(\mathbf{x}, \mathbf{y}_{1:n}) = \frac{\text{RNN}(\mathbf{y})}{\|\text{RNN}(\mathbf{y})\|_2} - \frac{1}{n} \sum_{i=1}^{n} \frac{e(y_i)}{\|e(y_i)\|_2} \tag{10}$$

where $\text{RNN}(\mathbf{x})$ is the final state of an RNN trained end-to-end with $s_{\text{voc}}$ using a discriminative loss:

$$L_{\text{voc}} = \sum_{(\mathbf{x}, \mathbf{y}_t) \in D, \mathbf{y}_r \sim D_{\mathbf{y}}} \log(2 + s_{\text{voc}}(\mathbf{x}, \mathbf{y}_t) - s_{\text{voc}}(\mathbf{x}, \mathbf{y}_r)). \tag{11}$$

During decoding, this score is combined with the language model *before* sampling next word candidates, while the other models are used to rescore candidates (see §3.4). The reason for this is that this model improves diversity in the next word distribution, while encouraging continuation of topics in the context.

### 3.2.6 LEXICAL STYLE MODEL

The goal of the lexical style model is to learn stylistic aspects of desirable writing based on observed lexical distributions. The scoring function is defined as

$$s_{\text{bow}}(\mathbf{y}) = \mathbf{w}_s^T \text{maxpool}(e(\mathbf{y})). \tag{12}$$

The model is trained with a similar ranking criteria as the relevance model, except that the negative examples are sampled from the language model:

$$L_{\text{bow}} = \sum_{\mathbf{y}_t \in D_{\mathbf{y}}, \mathbf{y}_s \sim P_{lm}(\mathbf{y}_s)} \log \sigma(s_{\text{bow}}(\mathbf{y}_t) - s_{\text{bow}}(\mathbf{y}_s)). \tag{13}$$

### 3.3 MIXTURE WEIGHT LEARNING

Once all the communication models have been trained, we learn the combined objective. In particular we learn the weight coefficients $\lambda_k$ in equation 1 to linearly combine the scoring functions, using a discriminative objective

$$L_{\text{mix}} = \sum_{(\mathbf{x}, \mathbf{y}) \in D} \log \sigma(f(\mathbf{x}, \mathbf{y}) - f(\mathbf{x}, \mathcal{A}(\mathbf{x}))), \tag{14}$$

where $\mathcal{A}$ is the inference algorithm for beam search decoding. The objective learns to rank the gold continuation above the continuation predicted by the current model. The full model is therefore trained discriminatively to classify sequences, and the same model is used to generate the sequences.

For each training iteration inference is performed based on the current values of $\lambda$. This has the effect that the objective function changes dynamically during training: As the current samples from the model are used to update the mixture weights, it creates its own learning signal by applying the generative model discriminatively.

**Data:** context $\mathbf{y} = y_1 \cdots y_n$, beam size $k$, mixture weights $\lambda$
**Result:** best continuation
best = None
beam = [**y**]
**for** *step = 0; step < max_steps; step = step +1* **do**
 next_beam = []
 **for** *candidate in beam* **do**
  expand(next_beam, next_k(candidate, $\lambda$))
  **if** *lookahead_score(candidate.append(termination_token)) > best.score* **then**
   best = candidate.append(term)
  **end**
 **end**
 **for** *candidate in next_beam* **do**
  candidate.score += $f_\lambda$ (candidate)            ▷ score with models
 **end**
 beam = sort(next_beam, key=score)[:k]          ▷ select top k candidates
**end**
**if** *learning* **then**
 update $lambda$ with gradient descent by comparing beam against the gold
**end**
return best

**Algorithm 1:** Inference/Learning in the Learning to Write Framework.

## 3.4 BEAM SEARCH

Due to the limitations of greedy decoding and the fact that our scoring functions do not decompose across time steps, we perform generation with a beam search procedure, shown in Algorithm 1. The naive approach would be to perform beam search based only on the language model, and then rescore the $k$ best candidate completions with our full model. We found that this approach leads to limited diversity in the beam and therefore cannot exploit the strengths of the full model.

Therefore we rescore the current hypotheses in the beam with the full (partial) objective function and keep the $k$ best candidate sequences after expanding to the next word. To further increase diversity we sample $k$ candidate next words from the distribution when expanding a hypothesis, instead of obtaining the $k$ highest scoring next words. In our experiments we generate text using a beam size of 8.

Two modules are handled differently during beam search: First, the working vocabulary score is integrated before sampling next word candidates, i.e. inside the `next_k` call. Second, the entailment score of a candidate is recomputed only when sentence-terminating punctuation is generated. Otherwise the current entailment score is re-used. Due to the nature of beam-search the entailment score is not accumulated across sentences; rather we assume that the effect of the last sentence score will already be reflected in the content of the beam when the next sentence is scored. While the first sentence is generated, $s'_{\text{entail}}$ takes an initial constant value to avoid bias towards incomplete sentences.

## 4 EXPERIMENTS

### 4.1 EXPERIMENTAL SETUP

**Corpora** We use two corpora for evaluation. The first is TripAdvisor reviews.[1] We use only reviews that have at least 5 sentences, using the first 4 as the context and the remainder as the ending to be generated. There are on average 11 sentences per review. We use the first 1M reviews for training. The second is the ROCStory corpus (Mostafazadeh et al., 2016), which is a collection of crowdsourced commonsense short stories, consisting of five sentences each (98k stories in the training set). We use this corpus to train our model to predict a coherent final sentence given the first four.

For the (larger) TripAdvisor corpus, we train the language model on that corpus alone. For the ROCStory corpus, we pretrain the model for 420M tokens on the Toronto Books corpus[2] before training on the ROCStory text.

---

[1]TripAdvisor corpus: `http://times.cs.uiuc.edu/~wang296/Data/`
[2]Toronto Books corpus: `http://yknzhu.wixsite.com/mbweb`

|  | **TripAdvisor** | **ROCStory** |
|---|---|---|
| CONTEXT | *Once you walk in the door, you feel the calm & peace in this one time monastery. Staff so welcoming and friendly. The garden area was an oasis to come back to and recharge before heading out into those magical streets. Buffet breakfast excellent.* | *I was ill-prepared for my final exam coming up. I decided to make a cheat sheet to use in class. I was nervous, but needed an A. I put my cheat sheet in my palm and used it.* |
| REFERENCE | *Would definitely return.* | *The teacher caught and suspended me.* |
| L2W | *Staff friendly and helpful. Rooms are spacious, clean and comfortable. Breakfast is excellent, with a wide selection of food, fresh fruit, juices, coffee, tea, hot chocolate.* | *After several minutes, the teacher caught me.* |
| RNN | *The rooms are spacious, clean and comfortable. The staff is very friendly and helpful. The location is perfect. It is within walking distance of the blue mosque, hagia sophia, topkapi palace, blue mosque, hagia sophia, topkapi palace, grand bazaar, spice bazaar, spice bazaar, spice bazaar, spice bazaar, spice bazaar, spice bazaar, spice bazaar, spice bazaar, spice bazaar, spice bazaar, spice bazaar, spice bazaar, spice bazaar, spice bazaar, spice bazaar, spice bazaar and.* | *I was very proud of myself for doing this.* |

Table 1: Examples of the true (human), learning2write, and RNN continuation of the same context.

**Models** We use the language model from Section 3.1 (a 2-layer RNN with 1024 GRU cells per layer) as our reference baseline. We include three variants of our model: **L2W (NO META-LEARN)** uses a uniform mixture of scores for the learned sub-objectives (no meta-weight learning); **L2W (NO WK. VOCAB)** omits the working vocabulary model (Section 3.2.5) but uses mixture weight learning for the other components; and **L2W (FULL)** is our full model.

To analyze the suitability of the reference endings and base language model samples to train the repetition model, we define a *repetition ratio* as the ratio of total words to unique words. We found that ROCStory endings generated by the language model have a higher repetition ratio than the reference endings (1.12 vs. 1.02). On TripAdvisor the difference is more pronounced, with a ratio of 1.48 for the language model against 1.17 for the references.

## 4.2 EVALUATION SETUP

Previous work has reported that automatic measures such as BLEU, ROUGE, and Meteor, do not lead to meaningful evaluation when used for long or creative text generation where there can be high variance among correct generation output (Wiseman et al., 2017b; Vedantam et al., 2015). For open-ended generation tasks such as our own, human evaluation is the only reliable measure (Li et al., 2016b; Wiseman et al., 2017b). However, we also report the automatic measures to echo the previous observation regarding the mismatch between automatic and human evaluation. Additionally we provide multiple generation samples that give insights into the characteristics of different models.

We pose the evaluation of our model as the task of generating an appropriate ending given an initial context of $n$ sentences. For the purposes of automatic evaluation, the ending is compared against a gold reference ending that was written by a human (drawn from the original corpus). For human evaluation, the ending is presented to a human, who assesses the text according to several criteria, which are closely inspired by Grice's Maxims:

1. *Quantity*: Does the generation avoid repetition?
2. *Quality*: Does the generation contradict itself (or the initial context)?
3. *Relation*: Does the generation appropriately continue the initial context?
4. *Manner*: Does the generation use the same style as the initial context?

Finally, a Turing test question is presented to each judge: was the ending written by a human?

For both automatic and human evaluation, we select 1000 passages from the test set of each corpus and generate endings using all models, using the initial $n = 4$ sentences as context. Automatic

**TripAdvisor**

| Algorithm | Automated | | | Human | | | | |
|---|---|---|---|---|---|---|---|---|
| | BLEU | ROUGE | Meteor | Quantity | Quality | Relation | Manner | Turing test |
| LANGUAGE MODEL | **24.11** | **18.35** | **16.90** | 2.45 | 3.59 | 3.15 | 2.91 | 38.70 |
| L2W (NO META-LEARN) | 0.00 | 10.04 | 9.48 | 4.01 | 3.88 | 3.40 | 3.36 | 60.20 |
| L2W (NO WK. VOCAB) | 0.00 | 12.76 | 10.48 | **4.62** | **4.46** | **4.07** | **3.80** | **75.10** |
| L2W (FULL) | 0.34 | 14.43 | 14.01 | 4.33 | 4.13 | 3.78 | 3.70 | 67.80 |
| REFERENCE | *100.00* | *100.00* | *100.00* | 4.52 | 4.35 | 4.27 | 4.16 | 90.70 |

**ROCStory**

| Algorithm | Automated | | | Human | | | | |
|---|---|---|---|---|---|---|---|---|
| | BLEU | ROUGE | Meteor | Quantity | Quality | Relation | Manner | Turing test |
| LANGUAGE MODEL | 15.22 | **9.52** | 5.53 | 4.30 | 3.21 | 2.60 | 3.50 | 51.90 |
| L2W (NO META-LEARN) | 2.67 | 4.00 | 0.78 | 2.22 | 2.56 | 1.98 | 1.95 | 19.20 |
| L2W (NO WK. VOCAB) | 7.54 | 6.24 | 2.94 | 2.69 | 2.75 | 2.28 | 2.47 | 26.20 |
| L2W (FULL) | **18.95** | 8.75 | **7.58** | **4.38** | **3.54** | **3.05** | **3.59** | **57.10** |
| REFERENCE | *100.00* | *100.00* | *100.00* | 4.68 | 4.63 | 4.53 | 4.39 | 88.70 |

Table 2: Scores of baselines and our model on both datasets. Top: TripAdvisor, bottom: ROCStory. Automated metrics are from 0–100. Human metrics are from 1–5, except for the Turing test, which is from 0–100. Each cell is the micro-averaged across 1000 datums from a held-out test set.

evaluation scores each generated ending against its reference ending. To conduct human evaluation, we present the initial context with the generation from each of the models individually to workers on Mechanical Turk, who score them using the above criteria. As a control, we also ask workers to score the original (human-written) reference endings for each selected passage.

## 5 RESULTS AND ANALYSIS

### 5.1 QUANTITATIVE ANALYSIS

Results for both automatic and human evaluation metrics are presented for both corpora in Table 2.

For the TripAdvisor corpus, the language model baseline does best on all of the automated metrics. However on all human metrics, the L2W (NO WK.VOCAB) model scores higher, and achieves nearly twice the percentage of passed Turing tests (75% compared to 39%). It scores even better than the L2W (FULL) model. Manual inspection reveals that for the TripAdvisor task, the topical focus and lexical specificity given by vocabulary scoring actually lead to slightly worse model behavior. In TripAdvisor reviews, many possible sentences could logically follow a reviews context. A well-behaving but generic language generator is better able to leave judge's unsurprised, if also uninformed. The full model brings additional focus to the generations, but at the cost of a reduction in overall coherence.

In the ROCStory generation task the language model is more competitive with L2W (FULL). The raw language model is able to take advantage of the simpler data inherent in this task: ROCStories are shorter (the endings are always a single sentence) and have a smaller vocabulary. However, the L2W full model still performs the best on all human metrics. The ROCStory corpus presents specific contexts that require a narrower set of responses to maintain coherency. The L2W (FULL) takes advantage of reranking by matching the topic of the context, yielding a higher 'Relation' score and more Turing test passes overall.

The very low BLEU scores observed in our results in the TripAdvisor domain are an artifact of the BLEU metrics length penalty. The average length of reference completions is 12 sentences, which is much longer than the average length of endings generated by our Learning to Write models. This forces the BLEU score's length penalty to drive down the scores, despite our observation that there is still a significant amount of word and phrase overlap. The completions generated by the base language model are longer on average (as it tends to repeat itself over and over) and therefore to not suffer from this problem.

### 5.2 QUALITATIVE ANALYSIS

*Learning to Write (L2W)* generations are more topical and coherently mold with the context. Table 1 shows that both the L2W system as well as the classic RNN start similarly, commenting on the hotel

staff and the room. L2W is able to condense the same content into fewer words, following the context's style. Beginning with the third sentence, L2W and the LM diverge more significantly. The LM makes a generic comment: *"The location is perfect,"* while L2W goes on to list specific items it enjoyed at breakfast, which was mentioned in the context. Furthermore, the LM becomes "stuck" repeating itself—a common problem for RNNs in general—whereas the L2W system finds a natural ending.

The L2W models do not fix every degenerate characteristic of RNNs. The TripAdvisor L2W generation in Table 1, while coherent and diverse, leaves room for improvement. The need to relate to topical phrases can override fluency, as with the long list of *"food, fresh fruit, juices, coffee, tea, hot chocolate."* Furthermore phrases such as *"friendly and helpful"* and *"clean and comfortable"* occur in the majority of generations. These phrases, while common, tend to have semantics that are expressed in many different surface forms in real writing (e.g., *"the staff were kind and quick to respond"*). Appendix A presents sample generations from all models on both corpora for further inspection.

## 6 RELATED WORK

**Generation with Long-term Context**  While relatively less studied, there have been several attempts at RNN-based paragraph generation for multiple domains including image captions (Krause et al., 2017), product reviews (Lipton et al., 2015; Dong et al., 2017), sport reports (Wiseman et al., 2017a), and recipes (Kiddon et al., 2016). Most of these approaches are based on sequence-to-sequence (seq-to-seq) type architectures. One effective way to avoid the degenerative characteristics of RNNsunder seq-to-seq is to augment the input with sufficient details and structure that can guide the output generation through attention and a copying mechanism. However, for many NLP generation tasks, it is not easy to obtain such the large-scale training data required to support a robust sequence-to-sequence model. For example, a recent dataset for image-to-paragraph generation contains 14,575 training pairs Krause et al. (2017), and state-of-the-art hierarchical models, while improving over strong baselines, still lead to generation that repeats, contradicts, and is generic.

**Alternative Decoding Objectives**  The tendency of RNN generation to be short, blend, repetitive and contradictory has been noted multiple times in prior literature. A number of papers propose approaches that can be categorized as *alternative decoding objectives* (Shao et al., 2017). For example, Li et al. (2016a) propose a *diversity-promoting objective* that interpolates the conditional probability score with negative marginal or reverse conditional probabilities. Unlike the negative marginal term that blindly penalizes generic generation, all our communication models are contextual in that they compare the context with continuation candidates. Incorporating the reverse conditional probability has been also proposed through the noisy channel models of Yu et al. (2017). The clear benefit of the reverse conditional probability model is that it can prevent the *explaining away* problem of the long-term context. However, decoding with the reverse conditional probability is significantly more expensive, making it impractical for paragraph generation. In addition, both above approaches do not allow for learning more expressive models than our work presents. The communication models presented in our work can be easily integrated into the existing beam search procedure, and each model is lightweight to compute.

The modified decoding objective has long been a common practice in statistical machine translation literature (Koehn et al., 2003; Och, 2003; Watanabe et al., 2007; Chiang et al., 2009). Notably, it still remains a common practice with neural machine translation, even with a highly expressive network structure trained on an extremely large amount of data (Wu et al., 2016). Inspired by all above approaches, our work presents a general learning framework together with a more comprehensive set of composite communication models. Unlike previous approaches, our composite models are trained to directly address the particular limitations of the base RNN models, by integrating RNN generation into the discriminative learning. This allows for customization of the decoding objective to better cope with the undesired behavior of the base language models.

**Meta Learning**  There has been a broad range of literature on meta learning, where the learning goal broadens the search space of learning by targeting to learn model architectures, hyperparameters, and learning algorithms that are typically hand-designed or selected. Andrychowicz et al. (2016), for example, learn the learning rate, while Zoph & Le (2016) train a neural network to gen-

erate neural network architectures. while Snoek et al. (2012) proposed Bayesian optimization to learn hyperparameters that cannot be optimized with gradient descent. The learning framework we presented here can be interpreted as a type of meta learning in a broad sense in that there are multiple layers of learning: learning the base RNN language model, then learning a collection of composite communication models that are customized to the particular behavior of the base model, and finally the generator that learns to combine all sub-components.

## 7 CONCLUSION

Our work presents a unified learning framework that can learn to generate long, coherent text overcoming the limitations of RNNs as text generation models. Our framework learns a decoding objective suitable for generation through a combination of sub-models that capture linguistically-motivated qualities of good writing. Our work makes a unique contribution that complements existing literature on long text generation that is predominantly based on seq-to-seq models with a large amount of in-domain training data; we demonstrate that the fluency of general RNN language models can be successfully guided to generate more lengthy and sensical text, which can be useful for domains where in-domain data is not sufficient to support seq-to-seq type training.

We propose a general framework for learning a decoding objective in two parts: learning component models to rank candidate generations, which are motivated by different aspects of Grice's Maxims, and learning a weighing scheme that balances the influence of each of these scoring functions. This framework is amenable to any number and kind of sub-objectives, allowing for multiple qualities of good writing to be modeled across diverse domains. Human evaluation shows that the quality of the text produced by our model exceeds that of RNN baselines by a large margin and the generations score significantly higher on a Turing test evaluation.

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

## A  EXAMPLE GENERATIONS

**ROCStory**

| | |
|---|---|
| **Context** | Gina was bored in detention. She decided to text her friend mary across the room. She knew the mean teacher wouldn't approve. She kept her phone in her purse to hide it from the teacher. |
| **Language Model** | She was so excited that she couldn't find her phone. |
| **No Meta-Learning** | Mary asked her mother if she could asked her if she was students. Teacher |
| **Learn2Write** | Mary was relieved when her friend came over. |
| **Human** | Mary texted back. |

| | |
|---|---|
| **Context** | Jack has a big garden. He weeds it every day. Once a week he takes veggies to the market. Jack waters his garden in the morning. |
| **Language Model** | He was so excited that he couldnt wait to go back to the store. |
| **No Meta-Learning** | Trees grow trees trees and fruit trees. Vegetables |
| **Learn2Write** | He loves his garden more than anything else. |
| **Human** | His veggies grow large and plentiful. |

| | |
|---|---|
| **Context** | Samantha wanted to watch a scary movie, but her mother said no. Her mother left for work, and samantha decided to watch it anyway. The movie was so scary it gave samantha nightmares for weeks! She wished shed never watched it! |
| **Language Model** | She was so excited that she couldnt sleep that night. |
| **No Meta-Learning** | She was so excited that she couldnt sleep that night. |
| **Learn2Write** | She vowed never to watch scary movies again. |
| **Human** | Samantha learned her lesson. |

| | |
|---|---|
| **Context** | My friend was moving away to college. We decided to throw her a party. We all met up with cake and snacks. She was so surprised to see us! |
| **Language Model** | She was so excited that she couldn't wait to go back. |
| **No Meta-Learning** | Birthday party dinner was celebration! Welcome |
| **Learn2Write** | We all hugged her goodbye and went home. |
| **Human** | She hugged everyone and cried in joy. |

**TripAdvisor**

| | |
|---|---|
| **Context** | The hotel staff were very kind and helpful. Our room was large but a little sparse. The garden is really nice in the courtyard. The location is perfect, next door to the station. |
| **Language Model** | If you want to be in the middle of the action, this is not the place for you. If you want to be in the middle of the action, this is not the place for you. If you want to be in the middle of the action, this is not the place for you. But if you want to be in the middle of the action, this is not the place for you. If you want to be in the middle of the action, you |
| **No Meta-Learning** | We didn't eat at the hotel so can't comment |
| **Learn2Write** | stay there again in a heartbeat. Catermeran is a great place to stay. |
| **Human** | There are no stairs to the hotel but quite a few in the hotel. Restaurants right outside the door. The only problem would be for people who are not early risers, the church bells go early ( about 7am ) are loud and go on for a while ( a few minutes ). We used it as an alarm, which was fine for us but some may not like it. The breakfast was a little adverage but no un-eatable. |

| | |
|---|---|
| **Context** | We spent 2 nights at hotel abbazia in late march with our teenage children. The location couldn't be better. . . . So near to the train station and the bus station. We also loved the fact that we were away from the crowds around st. marks square ( nice place to visit, but i wouldn't want to live there : ) the hotel provided a good breakfast and there were some great restaurants in the area. " La zucca " was our favorite but there was another that the hotel suggested ( right across and down from the hotel towards the canal. . . Can't remember the name ) that was also very good. |
| **Language Model** | The staff were very friendly and helpful. Breakfast was included in the price and was very good. Overall, i would recommend this hotel and would |
| **No Meta-Learning** | The staff was very friendly and helpful. The room was very clean and the bed was very comfortable. I would definitely stay here again and would recommend it to anyone who wants to stay in the heart of the city. I would recommend this hotel to anyone who wants to stay in the heart of the city. I would recommend this hotel to anyone who wants to be in the heart of the city. I would recommend this hotel to anyone who wants to be in the heart |
| **Learn2Write** | The room was clean and the bed was comfortable. |
| **Human** | We also loved the pizza at " antico forno " near the rialto bridge. The staff at abbazia are fantastic. . . Very friendly and very helpful. The rooms are more than adequate and very clean. We had a family room that didn't have a stand-up shower and that was probably the only minor negative i can think of ( a little tough for my 6' 4 " son to cram into the tub ) ps loved waking up to the church bells. . . . No alarm needed! |

| | |
|---|---|
| **Context** | After reading multiple reviews for liki tiki village i was a bit nervous about staying there. We loved every minute. I wish that we had more time to enjoy the pools, but we were visiting the parks most of the time. The rooms were clean and the kitchen well stocked. |
| **Language Model** | The staff was very friendly and helpful. I would recommend this hotel to anyone who is looking for a nice place to stay in the heart of the city. I would highly recommend this hotel to anyone who is looking for a place to stay in san diego. I would highly recommend this hotel to anyone who is looking for a place to stay in san diego. I would highly recommend this hotel to anyone who is looking for a place to stay in san diego. If you |
| **No Meta-Learning** | Our 6euros was multi-stop, vegatarian vegatarian vegatarian 6euros |
| **Learn2Write** | The rooms were clean and the kitchen well stocked. The staff were very friendly and helpfu! I would definately recommend this hotel to anyone who is looking for a place to stay in the heart of san diego. Thank you to all of the staff for making our stay so enjoyable! |
| **Human** | We did go to the time share presentation but we left almost immediately since we are owners elsewhere. I guess they figured out that they would be wasting their time. Great to have a washer and dryer. Also great to have a seperate room for the kids. The grounds were clean and we felt safe. Checkin was a little confusing since they gave us a map of the property and then wrote all over it in black pen so that you could not see the map. Overall it was a great experience. Close to the grocery store and to target. A little difficult to find but the gps got us there. Great for the price and amenities. |

| Context | The location is amazingly convenient. Just 3min walk from the bus station or 1min walk from the train station and less than a 100m from the canal grande. I liked the fact that it's at the start of the city so that you begin your venice experience slowly. I recommend walking to the san marco piazza on your first time ( make sure you wear comfortable shoes ). |
|---|---|
| Language Model | The hotel is in a great location. The room was clean and comfortable. The staff was very friendly and helpful. The breakfast was very good. The staff was very friendly and helpful. I would recommend this hotel to anyone who is looking for a nice place to stay in venice. I would recommend this hotel to anyone who is looking for a place to stay in venice. I would recommend this hotel to anyone who is looking for a place to stay while in |
| No Meta-Learning | The staff were very friendly and helpful and the breakfast buffet |
| Learn2Write | The rooms are spacious and clean. Breakfast is very good. |
| Human | Then you exit the alley and face the magnificent san marco piazza and trust me, you will be pleasantly shocked. What an experience!!! : ) ) the abbazia rooms are decent and very clean. Breakfast is poor but adequate and wi-fi is free. The garden is very peaceful and offers some very relaxing moments. I was worried about noises from the train station next door but you can't hear a thing so no problem there. The guys at the reception are amazing. Very friendly and very helpful : ) ) what you want from a hotel in venice is a decent place to sleep, have a relaxing bath and some breakfast in the morning. From then on you will be spending all your time in town anyway so fo me the abbazia hotel was an excellent choice and i will go back for sure. Price is not cheap, but nothing is cheap in venice anyway. |

## B  MODEL DETAILS AND HYPERPARAMETERS

**Base language model**   We use a 2-layer RNN language model with 1024 GRU cells per layer. Embeddings vectors are of length 1024. Following Inan et al. (2017) we tie the input and output embedding layers' parameters. To regularize we dropout (Srivastava et al., 2014) cells in the output layer of the first layer with probability 0.2. We use mini-batch stochastic gradient descent (SGD) and anneal the learning rate regularly when the validation set performance fails to improve. Learning rate, annealing rate, and batch size were tuned on the validation set for each dataset, with details in section **??**. Gradients are backpropagated 35 time steps and clipped to a maximum value of 0.25.

**Entailment model**   Dropout is performed on both the input word embeddings and the MLP hidden layer with rate 0.5. Training is performed with Adam (Kingma & Ba, 2015) with learning rate 0.0005, batch size 128.

**Relevance model**   The convolutional layer is a 1D convolution with filter size 3 and stride 1; the sequences are padded so that the sequence output is the same length as the input. The model is trained with Adam, learning rate 0.001 and dropout is applied before the final linear layer with rate 0.5.

**Meta-weight learning**   Training is performed with SGD with a learning rate of 1.

