# OpenReview forum: "Learning to Write by Learning the Objective"
_ICLR.cc/2018/Conference — Invite to Workshop Track_

### Official Review · AnonReviewer3 · 2017-11-27
**This paper combines RNN language model with several discriminatively trained models to improve the language generation. I like the idea of using Grice’s Maxims of communication to improve the language generation. However, some parts need to be further clarified and it would be nice to see more related analysis.**

**Rating:** 5
**Confidence:** 4

**Review:**

This paper argues that the objective of RNN is not expressive enough to capture the good generation quality. In order to address the problems of RNN in generating languages, this paper combines the RNN language model with several other discriminatively trained models, and the weight for each sub model is learned through beam search.

I like the idea of using Grice’s Maxims of communication to improve the language generation. Human evaluation shows significant improvement over the baseline. I have some detailed comments as follows:

- The repetition model uses the samples from the base RNNs as negative examples. More analysis is needed to show it is a good negative sampling method.

- As Section 3.2.3 introduced, “the unwanted entailment cases include repetitions and paraphrasing”. Does it mean the entailment model also handles repetition problem? Do we still need a separate repetition model? How about a separate paraphrasing model?

- Equation 6 and the related text are not very clearly represented. It would be better to add more intuition and better explained.

- In the Table 2, the automated bleu scores of L2W algorithm for Tripadvisor is very low (0.34 against 24.11). Is this normal? More explanation is needed here.

- For human judgement, how many scores does each example get? It would be better to get multiple workers on M-Turk to label the same example, and compute the mean and variance. One score per example may not be reliable.

- It would be interesting to see deeper analysis about how each model in the objectives influence the actual language generation.

---

> ### Author Response · Authors · 2018-01-05
> **Thank you for your concrete and constructive feedback.**
>
> We added an analysis to the paper of the frequency of repetitions in the training data, finding that they indeed occur more frequently in the samples from the language model, which are used as negative examples for training the repetition model, than in the reference endings.
>
> Entailment examples in our training data are often but not always a form of paraphrasing, but usually not instances of direct repetition. Therefore we believe that we still need a seperate repetition model to handle more direct repetitions at a lexical level. A separate paraphrasing model is an interesting suggestion for future work, although we believe that the repetition and entailment models together are able to capture most of the paraphrases we are aiming to avoid.
>
> We improved the description of the entailment score formulation (eq 6).
>
> The very low BLEU scores observed in our results in the TripAdvisor domain are an artifact of the BLEU metric’s length penalty. The average length of reference completions is 12 sentences, which is much longer than the average length of endings generated by our Learning to Write models. This forces the BLEU score's length penalty to drive down the scores, despite our observation that there is still a significant amount of word and phrase overlap. The completions generated by the base language model are longer on average (as it tends to repeat itself over and over) and therefore do not suffer from this problem.
>
> While we agree that more labels per example will be valuable, we believe that the the test sets (of 1000 examples per domain) are large enough that to obtain a reasonably accurate aggregate score, despite the fact that not all of the individual annotations will be reliable.

---

### Official Review · AnonReviewer2 · 2017-11-28
**Well-motivated goals, but the methods don't achieve them.**

**Rating:** 4
**Confidence:** 5

**Review:**

This paper proposes to improve RNN language model generation using augmented objectives inspired by Grice's maxims of communication. The idea is to combine the standard word-by-word decoding objective with additional objectives that reward sentences following these maxims. The proposed decoding objective is not new; reseachers in machine translation
 have worked on it referring to it as loss-augmented decoding: http://www.cs.cmu.edu/~nasmith/papers/gimpel+smith.naacl12.pdf
The use of RNNs in this context might be novel though.

Pros:
- Well-motivated and ambitious goals

- Human evaluation conducted on the outputs.

Cons:
- My main concern is that it is unclear whether the models introduced are indeed implementing the Gricean maxims. For eaxample, the repetition model would not only discourage the same word occurring twice, but also a similar word (according to the word vectors used) to follow another one.

- Similary, for the entailment model, what is an "obvious" entailment"? Not sure we have training data for this in particular. Also, entailment suggests textual cohesion, which is conducive to the relation maxim. If this kind of model is what we need, why not take a state-of-the-art model?

- The results seem to be inconsistent. The working vocabulary doesn't help in the tripAdvior experiment, while the RNN seems to work very well on the ROCstory data. While there might be good reasons for these, the point for me is that we cannot trust that the models added to the objective do what they are supposed to do.

- Are the negative examples generated for the repetition model checked that they contain repetitions? Shouldn't be difficult to do.

- Would be better to give the formula for the length model, the description is intuition but it is difficult to know exactly what the objective is

- In algorithm 1, it seems like we fix in advance the max length of the sentence (max-step).  Is this the case? If so why? Also, the proposed learning algorithm only learns how to mix pre-trained models, not sure I agree they learn the objective. It is more of an ensembling.

- As far as I can tell these ideas could have been more simply implemented by training a re-ranker to score the n-best outputs of the decoder. Why not try it? They are very popular in text generation tasks.

---

> ### Author Response · Authors · 2018-01-05
> **Thank you for your detailed feedback.**
>
> Regarding our approach to implementing Grice's maxims:
>
> - Our repetition module is trained to recognize both exact repetitions and repetitions involving lexical paraphrases, as indicated by the cosine similarity between word embeddings. We believe that this is a more robust approach than placing hard constraints on repetition, and has the advantage that the model can learn to distinguish between desirable and undesirable similarity patterns in human-produced and machine-produced text.
>
> - For the entailment module, while there is a risk that relevant sentences will be penalized, in the training data most of the entailments are direct enough that they are not likely to occur in writing, while the neural class training examples still often contains relevant information.
> In terms of the model, we chose to use a lightweight bag-of-words model for time and memory efficiency reasons (as it is expensive to do pairwise sentence comparisons to compute the entailment scores), even though a state-of-the-art model is likely to somewhat increase the performance.
>
> We added an analysis to the paper of the frequency of repetitions in the training data, finding that they indeed occur more frequently in the samples from the language model, which are used as negative examples for training the repetition model, than in the reference endings.
>
> We added an equation in description of the length module in order to clarify its objective.
>
> The purpose of the maximum length restriction is simply to guarantee that the beam-search will terminate. In practice the generated sequence (which is the highest-scoring sequence ending with the termination token) is always shorter than the maximum length allowed.
>
> As suggested, we include a reranker baseline in the results to re-score the n-best outputs after doing beam search decoding using only the language model: We found that it performs much worse due to a lack of diversity in the beam.
>
> We added more details in the paper to support the claim that the objective is being learned. The scoring function learned in one stage informs the objective in the following stages. First, the expert classifiers are learned to improve the language model by using samples from the language model as negative training data. Subsequently, these expert classifiers are combined in a mixed objective where the weights of the classifiers are learned discriminatively. As a result, the overall objective function for training the generator changes dynamically as the mixture weights are updated because the objective itself depends on those weights. The mixture weights are learned to optimize a discriminative objective, which updates the overall generation objective; this in turn changes the discriminative objective for the next training iteration.

---

> > ### Comment · AnonReviewer2 · 2018-01-11
> > **Response to author response**
> >
> > While the paper was improved, it didn't address my main concern, that it is unclear whether the model really implements Gricean maxims. Assessing the repetitions and finding that the language model repeats slightly more often is not much evidence in my opinion. Also, the re-ranker should be trained on appropriately generated data, as it happens with the approach proposed. Thus my assessment of the paper remains the same.

---

### Official Review · AnonReviewer1 · 2017-11-30
**Neat contribution that integrates previous work**

**Rating:** 6
**Confidence:** 5

**Review:**

This paper proposes to bring together multiple inductive biases that hope to correct for inconsistencies in sequence decoding. Building on previous works that utilize modified objectives to generate sequences, this work proposes to optimize for the parameters of a pre-defined combination of various sub-objectives. The human evaluation is straight-forward and meaningful to compensate for the well-known inaccuracies of automatic evaluation.

While the paper points out that they introduce multiple inductive biases that are useful to produce human-like sentences, it is not entirely correct that the objective is being learnt as claimed in portions of the paper. I would like this point to be clarified better in the paper.

I think showing results on grounded generation tasks like machine translation or image-captioning would make a stronger case for evaluating relevance. I would like to see comparisons on these tasks.

----
After reading the paper in detail again and the replies, I am downgrading my rating for this paper. While I really like the motivation and the evaluation proposed by this work, I believe that fixing the mismatch between the goals and the actual approach will make for a stronger work.

As pointed out by other reviewers, while the goals and evaluation seem to be more aligned with Gricean maxims, some components of the objective are confusing. For instance, the length penalty encourages longer sentences violating quantity, manner (be brief) and potentially relevance. Further, the repetition model address the issue of RNNs failing to capture long-term contextual dependencies -- how much does such a modified objective affect models with attention / hierarchical models is not clear from the formulation.

As pointed out in my initial review evaluation of relevance on the current task is not entirely convincing. A very wide variety of topics are feasible for a given context sentence. Grounded generation like MT / captioning would have been a more convincing evaluation. For example, Wu et al. (and other MT works) use a coverage term and this might be one of the indicators of relevance.

Finally, I am not entirely convinced by the update regarding "learning the objective". While I agree with the authors that the objective function is being dynamically updated, the qualities of good language is encoded manually using a wide variety of additional objectives and only the relative importance of each of them is learnt.

---

> ### Author Response · Authors · 2018-01-05
> **Thank you for your feedback and suggestions.**
>
>                                                                                                                                                                                                                                                                                                                                                                                                                                   We added more details in the paper to support the claim that the objective is being learned. The scoring function learned in one stage informs the objective in the following stages. First, the expert classifiers are learned to improve the language model by using samples from the language model as negative training data. Subsequently, these expert classifiers are combined in a mixed objective where the weights of the classifiers are learned discriminatively. As a result, the overall objective function for training the generator changes dynamically as the mixture weights are updated because the objective itself depends on those weights. The mixture weights are learned to optimize a discriminative objective, which updates the overall generation objective; this in turn changes the discriminative objective for the next training iteration.
>
> The recommendation to tackle grounded language tasks is a great suggestion, and we are eager to explore this avenue for future work. We believe incorporating grounding introduces novel challenges and so falls out of the scope of this paper, which we have scoped to focus on open-ended, ungrounded generation.

---

### Decision · Program_Chairs · 2018-01-29
**ICLR 2018 Conference Acceptance Decision**

**Decision:**

Invite to Workshop Track

**Comment:**

I (and some of the reviewers) find the general motivation quite interesting (operationalizing the Gricean maxims in order to improve language generation). However, we are not  convinced that the actual model encodes these maxims in a natural and proper way.  Without this motivation, the approach can be regarded as a set of heuristics which happen to be relatively effective on a couple of datasets.  In other words, the work seems too preliminary to be accepted at the conference.

Pros:
-- Interesting motivation (and potential impact on follow-up work)
-- Good results on a number of datasets
Cons:
-- The actual approach can be regarded as a set of heuristics, not necessarily following from the maxims
-- More serious evaluation needed (e.g., image captioning or MT) and potential better ways of encoding the maxims

It is suitable for the workshop track, as it is likely to stimulate an interesting discussion and more convincing follow-up work.